# The Study of Novel Self-Rewetting Fluid Application to Loop Heat Pipe

**Jhih-Huang Gao [1], Shen-Chun Wu [2,\*], Ya-Wei Lee [1], Ta-Li Chou [3] and Yan-Chun Chen [3]**

[1] Department of Mechatronic, Energy and Aerospace Engineering, Chung Cheng Institute of Technology, National Defense University, Taoyuan 334, Taiwan; gao19881213@gmail.com (J.-H.G.); yaweilee@ndu.edu.tw (Y.-W.L.)

[2] Department of Aviation Mechanical Engineering, China University of Science and Technology, Taipei 115, Taiwan

[3] Department of Mechanical Engineering, National Taiwan University, Taipei 106, Taiwan; trade98tw@gmail.com (T.-L.C.); avid1115@gmail.com (Y.-C.C.)

\* Correspondence: mimi1210@seed.net.tw; Tel.: +886-03-5935707

**Abstract:** The purpose of this paper is to develop SRF formulations for LHP performance enhancement. In this paper, the solute of SRF is prepared, and butanol and its isomer, 2-butanol, are selected. In this paper, concentrations of the 2-butanol aqueous solution (10%, 15%, and 20%) plus the butanol 6% aqueous solution were used to measure the surface tension of four different formulations of SRF and water. It was found that the higher the solute concentration became, the stronger the Marangoni effect was, and the more obvious the surface tension change of the 2-butanol 20% aqueous solution was. Water, the butanol 6% aqueous solution and the 2-butanol 20% aqueous solution were filled into LHP respectively, and the heat transfer performance was measured. The 2-Butanol 20% aqueous solution improved LHP performance by about three times compared with water, and the lowest total thermal resistance was only 1/4 that of water. Therefore, the 2-butanol 20% SRF aqueous solution is an ideal formula for improving the LHP heat transfer performance.

**Keywords:** 2-butanol; self-rewetting fluid (SRF); loop heat pipe (LHP); formula

## 1. Introduction

In 1865, the following was proposed by Italian physicist Carlo Marangoni [1]: when two liquids are in contact, the liquid with strong surface tension will pull the liquid with weak surface tension a phenomenon later academically named the Marangoni effect.

The working fluids often used as two-phase heat transfer devices, such as pure water, ethanol, acetone, etc., are generally pure material fluids. Figure 1 shows that the surface tension of ordinary fluids (ordinary liquid) will gradually decrease with the increase in operating temperature. In 1973, scholars Vochten and Petre [2], and in 2004, scholars Abe et al. [3] measured the surface tension of different alcohol aqueous solutions, and found that alcohol aqueous solutions with high carbon chain (carbon chain number $\geq$ 4) have a different surface tension curve, such as the self-rewetting fluid of the red line in Figure 1. In particular, when the temperature increases to the β point, the surface tension of the aqueous solution will increase with the temperature. With an increase to the γ point domain, it can be observed that the (β-γ) domain has a reverse gradient of surface tension that is reversed by the temperature rise. Fluids with this characteristic are different from ordinary fluids and are called self-rewetting fluids (SRF).

When the self-rewetting fluid enters the reverse surface tension gradient region (β-γ), due to the Marangoni effect, it means that the fluid at the vapor–liquid two-phase interface will flow from the place with low surface tension to the place with high surface tension, and the temperature will be lower. The liquid is brought to a place with a higher temperature

and thus wets the surface with a high temperature, which has to delay the occurrence of drying out and enhance the heat transfer efficiency.

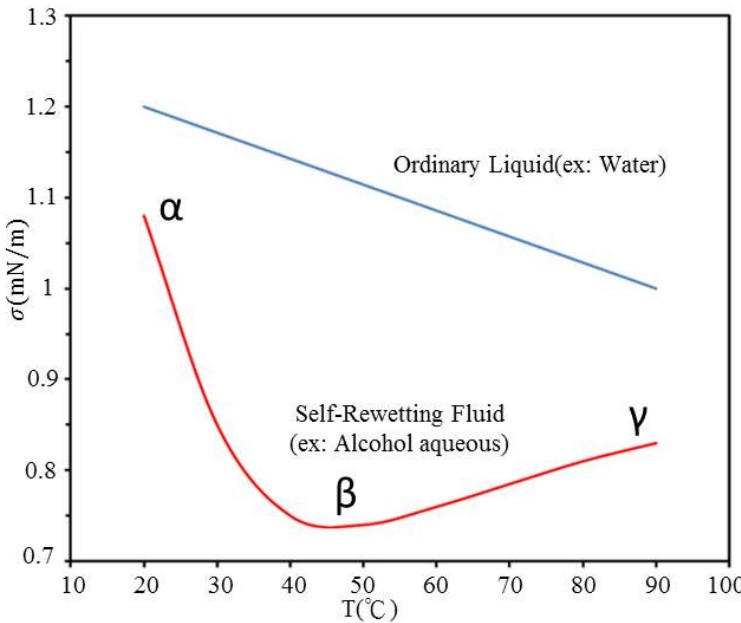

**Figure 1.** Identification of self-rewetting fluid [3].

The working fluid in this research is intended to be a self-rewetting fluid, which has better wettability than the pure material working fluid. The relationship between concentration, temperature and surface tension is shown in Equation (1):

$$\frac{\partial \sigma}{\partial X} = \sigma_T \frac{\partial T}{\partial X} + \sigma_C \frac{\partial C}{\partial X} \tag{1}$$

where $\sigma$ is the surface tension, $T$ is the temperature of the working fluid, and $C$ is the concentration of the working fluid.

Usually, the working fluids of the two-phase heat transfer device, such as pure water, ethanol, acetone, etc., are all pure substances, and their surface tension will gradually decrease with the increase in the operating temperature. In this paper, SRF is applied to the loop heat pipe (LHP), and the related literature is reviewed as follows.

In 2001, Zhang et al. [4] pointed out that both traditional heat pipes and loop heat pipes are faced with a phenomenon. The surface tension of the working fluid decreases with the increase in temperature, which is unfavorable for the extension or rewetting of the heated surface liquid. Furthermore, they suggested that the best way to solve this problem is to use a working fluid whose surface tension increases with temperature.

In 2006, Abe [5] sorted out the research over the years and pointed out that the self-rewetting fluid can push the cooler liquid to the hotter surface through the Marangoni effect because of the reverse surface tension gradient, delay the occurrence of drying, and then improve the heat transfer. The performance is helpful for the heat transfer performance of a non-capillary structure heat pipe, general heat pipe and boiling phenomenon.

In 2008, Francescantonio et al. [6] applied the self-rewetting fluid of the 0.1% heptanol aqueous solution to the heat pipe. The experimental results showed that the maximum heat load was more than twice that of water, which not only reduced the thermal resistance, but also delayed the occurrence of dry-out.

In 2011, Morovati et al. [7] used the butanol aqueous solution as the working fluid (the concentration range included 2%, 5%, and 7%), and carried out the pool boiling heat transfer experiment. The flux increased substantially, and the heat flux did not have a significant effect when the concentration increased to 7%.

In 2011, Launy et al. [8] sorted out the literature on the theoretical prediction and parameter discussion of loop heat pipes over the years. The paper pointed out the selection and filling amount of the working fluid, the material of the capillary structure, the parameters and geometric dimensions, the temperature and the heat sink temperature. Ambient temperature, the design of evaporator and compensation chamber, the inclination of system operation, and the presence of non-condensable gas will all affect the performance of the loop heat pipe. Among them, the capillary structure parameters have the most important influence on the performance of the loop heat pipe.

In 2014, Hu et al. [9] used the SRF fluid in micro oscillating heat pipes (MOHPs), and the SRF fluid of heptanol 0.1% aqueous solution and deionized water were used as working fluids to carry out the heat transfer experiments. The result shows that SRF has smaller surface tension than deionized water, which is beneficial to improve the efficiency and improve the heat transfer performance.

In 2015, Wu [10] first reported that SRF fluid was applied to LHP with a Ni wick to measure the surface tension of SRF (including butanol, pentanol and hexanol aqueous solutions) with different concentrations of alcohol aqueous solutions. The Marangoni effect becomes more vigorous, but above the saturation concentrations, the effect does not increase any further. Therefore, the optimal concentration of SRF is the saturation concentration, and the surface tension measurement results show that the critical heat load is increased by 1.6 times, and the total thermal resistance is reduced by about 60% compared with the heat transfer performance of the water working fluid.

In 2017, Wu et al. [11] first reported the application of SRF to LHP of the PTFE wick. The PTFE wick has a low thermal conductivity and can overcome heat leakage during LHP operation. However, PTFE is a hydrophobic material and cannot use pure water, the traditional working fluid. The butanol aqueous solution of SRF successfully passed the PTFE wick. After the LHP performance test, it was found that the butanol 6% aqueous solution PTFE system reduced the total thermal resistance by half, and the heat flux increased by more than 50% compared with the water–nickel system.

In 2017, Naresh et al. [12] applied the SRF of aqueous solutions of different alcohols (butanol, pentanol, hexanol, and heptanol) to thermosyphon, and also pointed out that the optimal concentration of SRF for different alcohol aqueous solutions is its saturation concentration.

In 2018, Hu et al. [13] used two SRFs, ethanol 5% and butanol 5% plus water, for the study of pool boiling. It was found that the critical heat flux (CHF) increase in the butanol 5% aqueous solution was 1.91 times higher than that of water due to the Marangoni effect caused by the surface tension gradient. It was pointed out that the Marangoni effect should be the key factor to enhance the heat transfer of pool boiling.

In 2019, Boubaker et al. [14] soaked the wick itself with the SRF butanol 3% aqueous solution and pure water, respectively, and conducted evaporation visualization experiments. It was found that with the increase in applied power, the vapor film formed by the wick when the working medium was pure water also changed. However, the shape of the vapor film formed by SRF decreases with the increase in the applied power, which more effectively solves the obstacle of the vapor film.

In 2021, Zaaroura et al. [15] applied the mixed fluid of the butanol 3% aqueous solution and gold nanofluid to the capillary heat pipe (CHP), pointing out that the butanol aqueous solution using SRF can reduce the evaporator temperature. Using visualization to observe the internal phase change, it was found that SRF can improve the vapor film barrier and effectively improve the heat transfer performance.

Summarizing the above literature, it is pointed out that the more obvious the Marangoni effect of SRF, the better its performance in the evaporative heat transfer operation. It is also found in the literature that the increase in the solute concentration of SRF [10,11] will increase the Marangoni effect and improve the LHP heat. Therefore, the purpose of this study is to develop an ideal SRF formulation for the improvement of heat transfer performance of LHP.

## 2. Materials and Methods

In this paper, the experimental method was firstly formulated and selected for the working fluid, then the aqueous solution of 2-butanol (10%, 15%, and 20%) was prepared, and the surface tension was measured with the aqueous solution of butanol 6% to confirm the SRF characteristics. The LHP system was firstly made with the Ni wick, and then the wick parameters were tested. Finally, the wick was placed into the LHP test system. The SRF, which was tested for surface tension and had an obvious Marangoni effect, was selected for the LHP performance test.

### 2.1. Working Fluid Choices

SRF formulations have the potential to enhance the heat transfer performance of LHP, so this paper focuses on finding the ideal SRF formulations to enhance the thermal transfer performance. In the literature [10–12], butanol, pentanol, hexanol, and heptanol aqueous solutions are shown to have the potential to enhance the heat transfer performance, but it is also mentioned that the higher the concentration, the better the heat transfer performance, among which the butanol 6% aqueous solution has the highest concentration. This paper searched for SRF with higher solubility, and the results of this analysis showed that 2-butanol, an isomer of Butanol, has a higher saturation concentration. In this experiment, it was used 2-butanol (10%, 15%, and 20%) as the formulation of SRF and the butanol 6% aqueous solution, and a surface tension measurement with temperature was conducted. The surface tension measurements were carried out with the temperature variation to measure whether the SRF has the characteristics of LHP heat transfer performance enhancement as described in the literature.

### 2.2. Surface Tension Measurement

The surface tension measurement experiment is shown in Figure 2. During the experiment, the DST-60 flat surface tension meter was used for measurement, and Ni was used as the substrate to observe the surface tension variation in different SRF. The substrate was dipped into the test working fluid (water, butanol 6% and 2-butanol (10%, 15%, and 20%)), and the specimen was subjected to surface tension. When the surface tension ($\sigma$) is balanced with the other forces, the upward tension *F*, the wetting length *L* and the contact angle $\theta$ between the specimen and the working fluid are recorded. Each sample was measured three times and the average was taken.

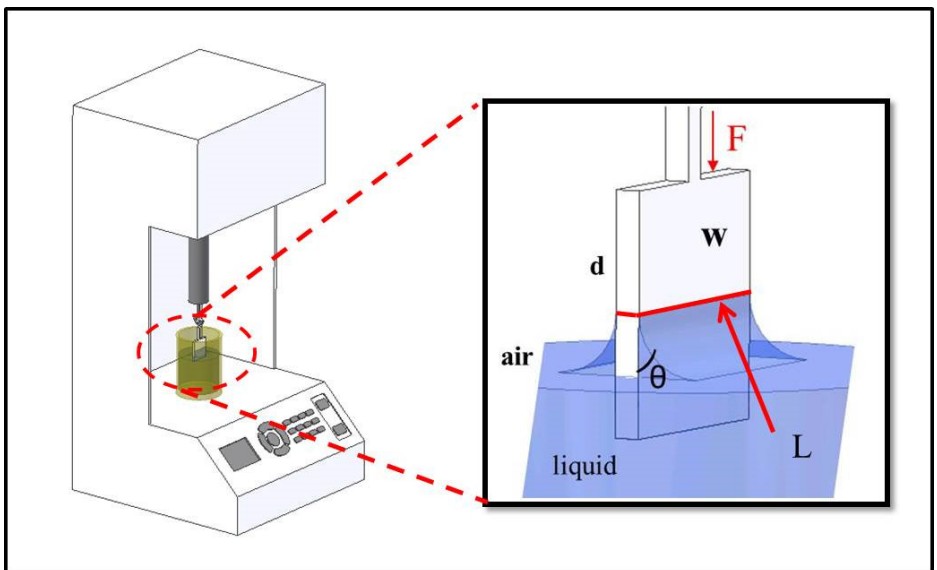

**Figure 2.** Surface tension measurement.

During the measurement process, the temperature of the working fluid is controlled by a constant temperature water bath, and the surface tension is measured at a fixed temperature (20, 40, 50, 60, and 70 °C) to observe the curve of the surface tension with the change in temperature and to judge the surface tension reverse phenomenon.

The surface tension value is calculated by substituting the measured data into Equation (2).

$$\sigma = \frac{F}{L \cdot \cos\theta} \tag{2}$$

### 2.3. Wick Manufacture

The manufacturing process of the Ni wick is shown in Figure 3. Type 255 nickel powder (2.2~2.8 µm) was selected as the wick material according to Tracey [16]. This Ni powder has a three-dimensional dendritic structure, which provides a high capillary force and good permeability. In this experiment, the loose filling method [17] was used to manufacture the Ni wick, in which the nickel powder is filled into the sintering mold without applying pressure to ensure the porosity of the capillary structure. The sintering temperature of Ni powder is set to increase to 625 °C within 60 min and hold the sintering temperature for 30 min. Hydrogen is passed through the sintering process for oxidation reduction. After the sintering is completed and the wick cools naturally, it can be removed from the mold to complete the wick production.

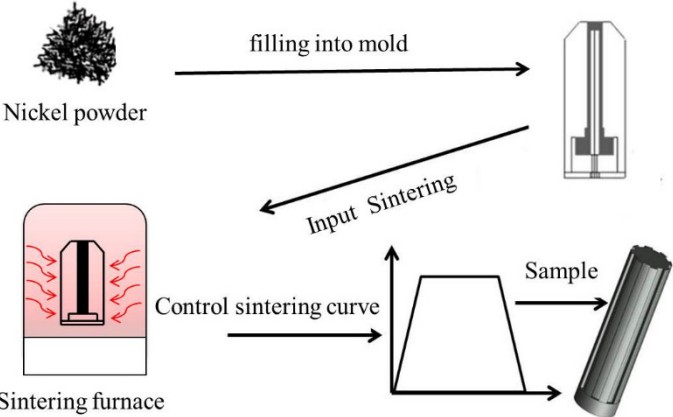

**Figure 3.** Wick manufacture process.

### 2.4. Wick Parameter Measurement

The parameters of the completed wick were measured using pure water as the working fluid and a self-designed test platform with reference to ASTM E128-99 [18] (see Figure 4). This platform measures important parameters, such as the effect radius ($r_c$), permeability ($K_w$), and porosity ($\varepsilon$). The effective radius is related to the size of the wick capillary force, while the permeability is related to the resistance of the working fluid to flow in the wick, and the porosity represents the number of effective pores that can generate the capillary force.

When the air pressure exceeds the capillary pressure that can be generated by the wick itself, the air will penetrate through the wick and float out of the water in the form of air bubbles, as follows in Equation (3).

$$r_c = \frac{2\sigma}{\Delta P_c} \tag{3}$$

where $\sigma$ is the surface tension value of the working fluid, and $\Delta P_c$ is the pressure difference corresponding to the first point of air bubble generation.

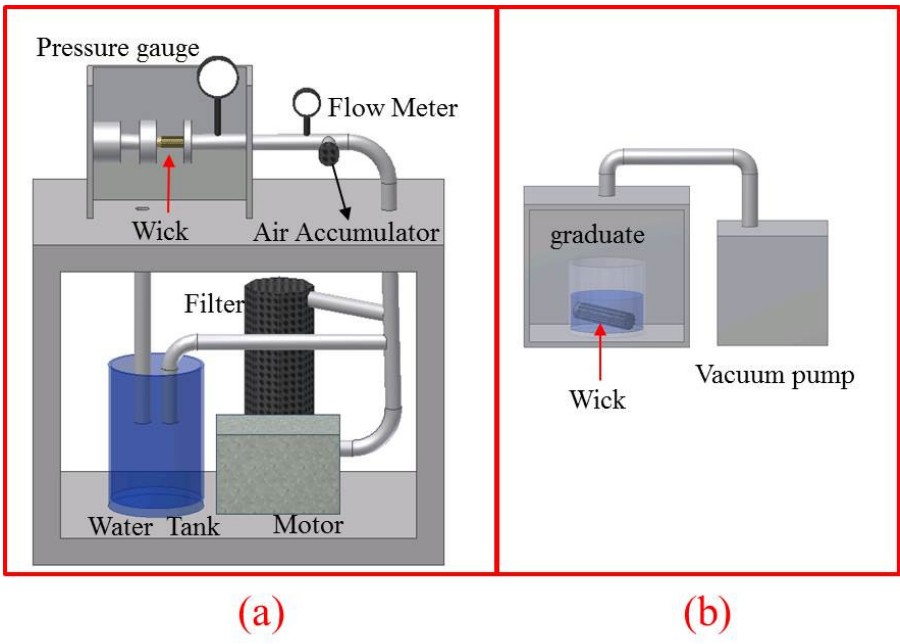

**Figure 4.** (**a**) Wick parameter measurement platform. (**b**) Porosity measurement.

The measured parameters are brought into Equation (4) to calculate the permeability parameters.

$$K_w = \frac{\dot{m}\mu_v}{2\pi\rho_v\Delta PL_w}\ln(T_w) \tag{4}$$

$L_w$ is the wick length; $\mu_v$ is the liquid viscosity coefficient; $\dot{m}$ is the mass flow rate; $\rho_v$ is the liquid density; $\Delta P$ is the pressure difference through both ends of the wick; and $T_w$ is the wick thickness.

Porosity measurements are mainly performed using this system (e.g., Figure 4b), where the wick is first placed in a container with a working fluid, then vacuumed to ensure the complete wetting of the wick, and the porosity is calculated as Equation (5). Water is used as the test fluid and measured using the Archimedes method.

$$\varepsilon = \frac{V_{pore}}{V_{total}} \tag{5}$$

where $\varepsilon$ is the porosity, $V_{pore}$ is the volume of the wick absorbing the test fluid, and $V_{total}$ is the volume of the wick.

In addition, in order to verify the pore characteristics of the wick, SEM images were taken to observe the pore structure under microscopy.

### 2.5. LHP Heat Performance Test

The parameters of the LHP test system are shown in Table 1. In the design of LHP, the basic requirements, such as heating area, space requirement, cooling target wattage, etc., as well as safety should be considered. Additionally, the materials of each component of the system, such as the working fluid, evaporator and other component materials, piping materials, wick materials, etc., should be selected to meet the requirements [10,19]. We should consider the space volume as a factor, then design the external dimensions of the components. In the heat transfer experiment, the Ni wick was used for the heat transfer test, and then aqueous solutions of water, butanol 6% and 2-butanol 20% were used as the working fluid.

**Table 1.** LHP system parameters.

| Working Fluid | Self-Rewetting Fluids |
|---|---|
| Heat load (W) | 25~500 |
| Ambient temperature (°C) | 20 |
| Evaporator | |
| Total length (mm) | 65 |
| Outer/inner diameter (mm) | 15.5/12.5 |
| Active length (mm) | 40 |
| Compensation chamber volume (cm$^3$) | 57 |
| Vapor line | |
| Outer/inner diameter (mm) | 6/4 |
| Length (mm) | 470 |
| Liquid Line | |
| Outer/inner diameter (mm) | 6/4 |
| Length (mm) | 583 |
| Condenser | |
| Outer/inner diameter (mm) | 6/4 |
| Length (mm) | 800 |
| Cooling water flow rate (L/min) | 4 |
| Heat sink temperature (°C) | 10 |

Figure 5 is the schematic diagram of the heat transfer performance test experiment. During the performance test, the thermocouple (T-Type) was arranged to measure the temperature of each part of the LHP, and the measurement points of each part were arranged as shown in the figure. The total thermal resistance value ($R_{LHP}$) is obtained by bringing the condenser inlet ($T_{c,in}$), evaporator wall temperature ($T_{evap}$), and power supply wattage (Q) into Equation (6). Meanwhile, Ref. [10] pointed out that SRF has better heat transfer performance at higher operating temperatures with the need for high power density heat dissipation components, such as solar energy storage in the energy field according to Gasia et al. [20]. Therefore, this experimental performance test continues to increase the heat load (Q) until LHP system performance failure, that is, the LHP system test stops.

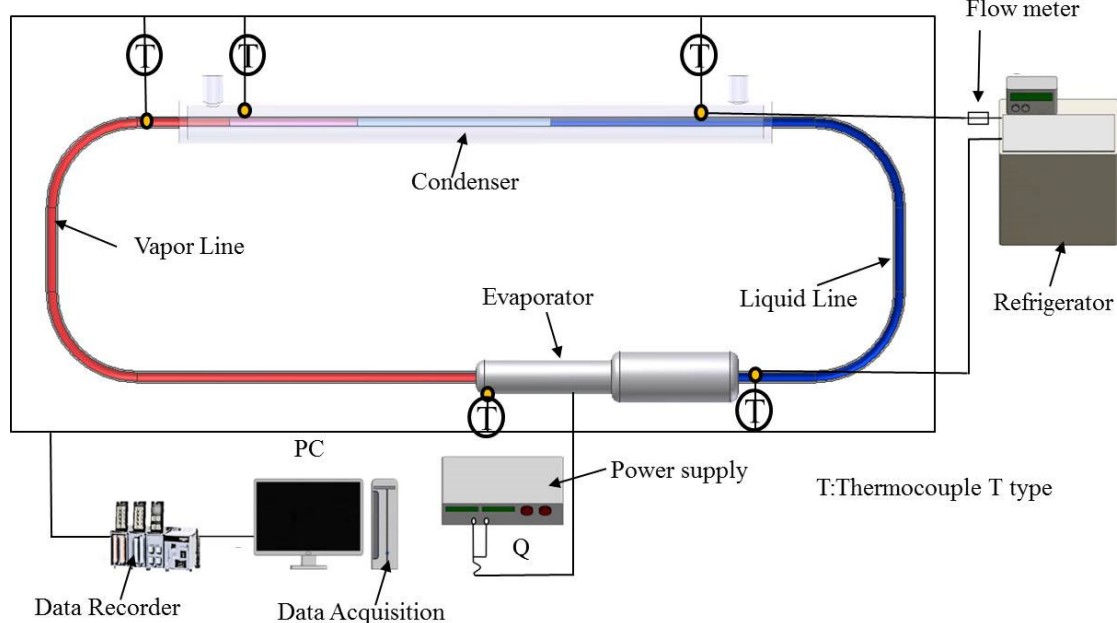

**Figure 5.** LHP heat transfer performance testing system setup.

In this paper, the performance of LHP is measured by the relative relationship between temperature–wattage (T − Q) and thermal-resistance–wattage (R − Q) to determine the performance.

$$R_{LHP} = \frac{T_{evap} - T_c}{Q} \tag{6}$$

Water, butanol 6% and 2-butanol 20% are tested as the working fluid. The ambient temperature is 25 °C, and the circulating water temperature of the condenser is 10 °C. The error of the thermocouple is ±0.2 °C. The error analysis is based on the relative inaccuracy analysis proposed by Moffat [21], and the error of the total thermal resistance of the system is ±1.9–16.2%.

## 3. Results and Discussion

### 3.1. Surface Tension Measurement Results

The surface tension of SRF (2-butanol and butanol 6% aqueous solution) was measured, and the surface tension of SRF decreased with the increase in temperature. Until the temperature reaches a specific temperature (40~50 °C), the surface tension of SRF increases with temperature, which is characteristic of SRF fluids. For 2-butanol 20%, the surface tension begins its reversal at about 40 °C; the surface tension value is about 19 mN/m; and the surface tension value is 48.5 mN/m when the temperature is 70 °C. In the temperature range of 40~70 °C, the surface tension measurement is about 20.5 mN/m for the 2-butanol 15% aqueous solution, and the surface tension measurement is about 42 mN/m at a temperature of about 70 °C. In the temperature range of 40–70 °C, the difference in surface tension is about 22.5 mN/m. The surface tension measurement is about 20.5 mN/m for the 2-butanol 15% aqueous solution, and the surface tension measurement is about 42 mN/m at a temperature of about 70 °C, or the 2-butanol 10% aqueous solution, the surface tension measurement is about 21.5 mN/m at a turning temperature of about 40 °C, and the surface tension measurement is about 37 mN/m at a temperature of about 70 °C. In the temperature range of 40–70 °C, the surface tension difference is about 16 mN/m. It can be seen that the higher the solute concentration of 2-butanol SRF, the larger the surface tension inverse gradient. In particular, the 2-butanol 20% inverse gradient is the most obvious, and in Ref. [11], it is stated that the higher the concentration of butanol, the more vigorous the Marangoni effect; the more vigorous Marangoni effect also raises the critical heat load.

In Figure 6, it can be observed that the higher the saturation concentration of SRF, such as the 2-butanol 20% aqueous solution, the lower the initial 20 °C surface tension value. It helps to evaporate the surface tension inverse gradient between 40 and 70 °C, and the size is also positively correlated with the SRF formulation concentration size. In the literature, Abe [3] stated that the larger the inverse gradient of SRF, the more pronounced the Marangoni effect.

In this paper, the surface tension was measured according to the saturation concentration of Butanol 6%, and the results showed that the surface tension value was about 31 mN/m at the turning temperature of the Marangoni effect of 50 °C, and about 46 mN/m at the temperature of 70 °C. In the temperature range of 50~70 °C, the surface tension difference was about 15 mN/m. Additionally, 15 mN/m is also the characteristic value of the Marangoni effect, and the larger the value, the more vigorous the Marangoni effect.

The results of Figure 6 show that the higher the concentration of the SRF formulation, the more obvious the Marangoni effect, but this is limited by the saturation concentration of the solute. Therefore, the SRF of each solute saturation concentration working fluid was selected, including the 2-butanol 20% aqueous solution, butanol 6% aqueous solution and non-SRF water, and we further conducted the heat transfer performance test of the LHP transfer performance test.

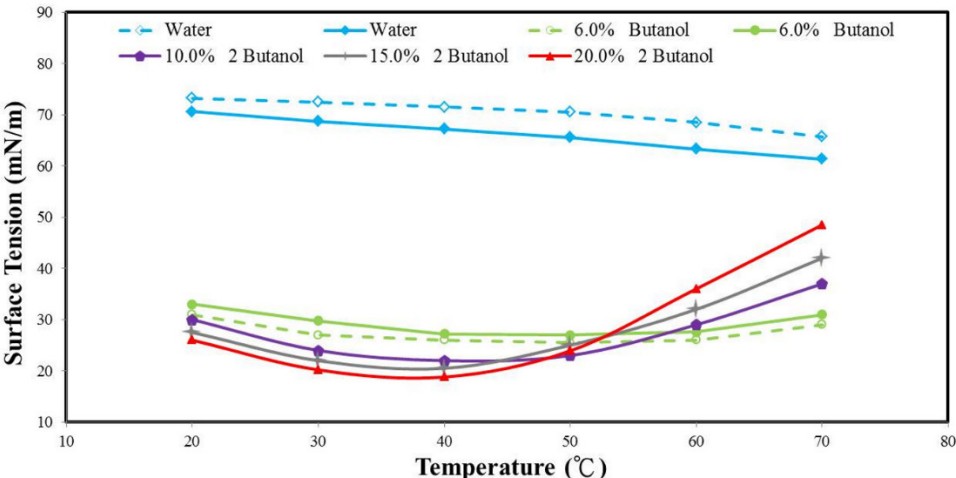

**Figure 6.** Surface tensions of different concentrations of aqueous solution as functions of temperature [11].

### 3.2. Wick Parameter

Figure 7 shows the finished Ni wick produced in this paper. In Figure 7a it can be observed that the Ni wick has a complete and good structural shape, and obviously has the characteristics of a steam groove design. The SEM of the Ni wick at 50 magnification is shown in Figure 7b, which shows that the Ni powder is closely arranged and has pore structural features. The pore size between the powder particles in Figure 7c is observed to be approximately between 2 and 20 μm, which provides sufficient pore capillary force.

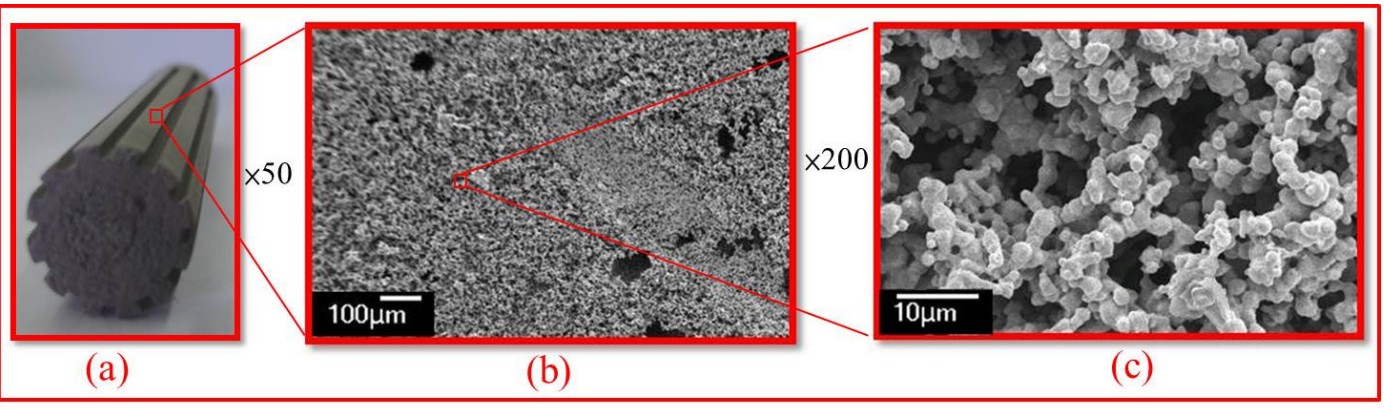

**Figure 7.** (**a**) Ni wick, (**b**) surface detail ×50, (**c**) SEM image ×1000.

The wick parameters is shown as Table 2. It can found that the wick effective pore radius $r_c$ (μm) is 2.4 (μm), the permeability $K_w$ ($10^{-13}$ m²) is 2.9 and the porosity $\varepsilon$ (%) is 64. These results are within the suggested range of Ref. [19], and their parameters are in accordance with [17], which means that this paper successfully produced the wick, and the property of the same wick is in line with the basic requirements of the wick for LHP operation.

**Table 2.** Wick internal parameters measurement results.

| Wick | Ni |
|---|---|
| Effective pore radius $r_c$ (μm) | 2.4 |
| Permeability $K_w$ ($10^{-13}$ m²) | 2.9 |
| Porosity $\varepsilon$ (%) | 64 |

*3.3. LHP Heat Performance*

To investigate the SRF in LHP, the dry-out was delayed, and the performance was enhanced by the Marangoni effect.

From Figure 8, it was found that the operating temperatures of all three working fluids increased as the thermal load increased. When increasing the thermal load, the slope of the evaporator wall temperature increase is the largest for water, followed by the butanol 6% aqueous solution, and the smallest is the 2-butanol 20% aqueous solution. This also shows that the overall operating temperature of the 2-butanol 20% aqueous solution is lower than the other two working fluids at different heat loads, which symbolizes that the application of the 2-butanol 20% aqueous solution to LHP has better heat transfer performance, and not only lower thermal resistance, but also lower operating temperature, which effectively improves the overall performance. The heat load of the 2-butanol 20% aqueous solution was operated up to 900 W. Compared with water and butanol, the drying phenomenon occurred at 250 and 650 W, respectively. In comparison with the results of Wu [11] (water and butanol 6%) whose performance was based on the operating temperature of 170 °C, the performance of water was 250 W and the performance of the butanol 6% aqueous solution was 600 W, which is consistent with the results of this paper. The performance of the 2-butanol 20% aqueous solution was about 2.4 times higher than that of water at 170 °C. The performance of the 2-butanol 20% aqueous solution was about 30% higher than that of Butanol 6% aqueous solution.

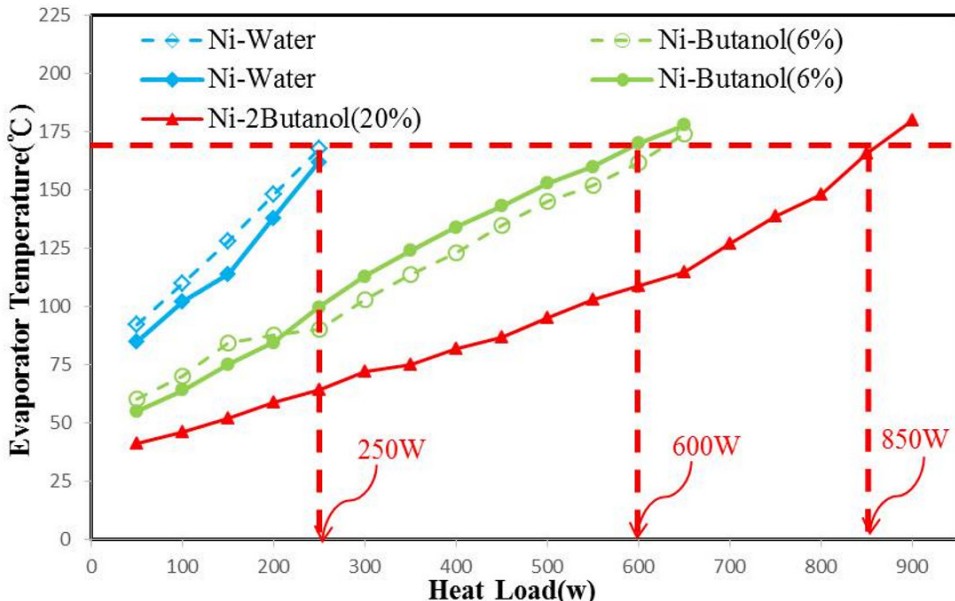

**Figure 8.** Evaporator temperature as a function of heat load for three different test samples [10].

The relationship between the thermal resistance and heat load is shown in Figure 9. According to the thermal resistance change with heat load, 200 W is used as the cut-off point in the graph. In the part of the area where the heat load is less than 200 W, there is a characteristic that the thermal resistance decreases with the heat load, which is called the variable thermal resistance area. Therefore, the diagram clearly distinguishes between A and B thermal resistance characteristics, which is a typical feature of LHP. Figure 9 shows that the thermal resistance of the fluid is highest at lower heat load before the LHP starts smoothly. After the heat load increases, the thermal resistance decreases until about 200 W enters the B region, where the heat load increases, but the thermal resistance of the system does not change much. The minimum thermal resistance of LHP with the 2-butanol 20% aqueous solution and water is 0.25 °C/W. In LHP operation with the 2-butanol 20% aqueous solution, the minimum total thermal resistance is about 0.16 °C/W. The minimum thermal resistance of the butanol 6% aqueous solution and water is 0.25 °C/W and 0.624 °C/W,

respectively. In addition, Figure 8 shows that the operating temperature of LHP with the 2-butanol 20% aqueous solution is always lower than that of LHP with butanol and water. The operating temperature of LHP with the 2-butanol 20% aqueous solution is always lower than that with butanol and water. The maximum heat load in Figure 9 is 900 W for the 2-butanol 20% aqueous solution, 650 W for the butanol 6% aqueous solution and 250 W for water. The 2-butanol 20% aqueous solution improves the performance of LHP by about three times compared with water, and the lowest total thermal resistance is only 1/4 of that of water as the working fluid, which is significantly reduced, so the 2-butanol 20% aqueous solution is the ideal formula to improve the performance of LHP. Therefore, the 2-butanol 20% aqueous solution is the ideal formula to improve the performance of LHP.

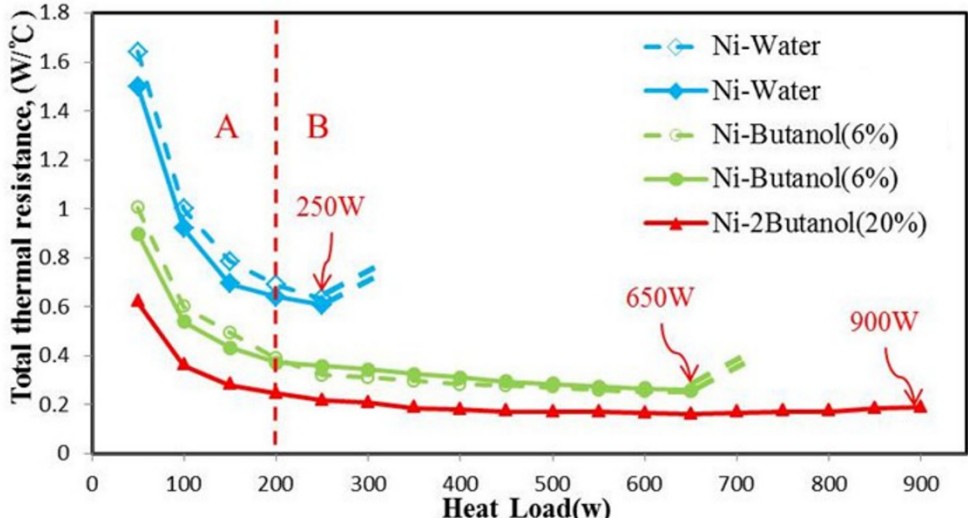

**Figure 9.** Thermal resistance as a function of heat load for three different test samples [10].

## 4. Conclusions

The purpose of this paper is to find the best SRF formulation for LHP application to enhance the performance of LHP. The conclusion is based on the experimental results of this paper as follows:

1.  The aqueous solution concentration and Marangoni effect showed a significant positive correlation. The order of exuberance was the 2-butanol 20% aqueous solution, 2-butanol 15% aqueous solution, 2-butanol 10% aqueous solution, and butanol 6% aqueous solution; in particular, the SRF effect of the 2-butanol 20% aqueous solution was the most obvious feature.
2.  The experimental results show that the maximum heat load with pure water is 250 W, and the maximum heat load with butanol 6% is 650 W. The maximum heat load with 2-butanol 20% is 900 W. The thermal resistance with 2-butanol 20% is reduced to 1/4, compared with water as the working fluid, and the heat load is increased by about three times.
3.  The higher the concentration of the 2-butanol aqueous solution, the more obvious the typical characteristics of SRF found in this paper. Therefore, the SRF formed by the 2-butanol 20% aqueous solution is the best formulation for LHP with the best performance.

**Author Contributions:** Conceptualization, J.-H.G. and S.-C.W.; data curation, T.-L.C. and Y.-C.C.; formal analysis, J.-H.G. and Y.-W.L.; investigation, J.-H.G. and T.-L.C.; methodology, J.-H.G. and S.-C.W.; resources, S.-C.W.; supervision, S.-C.W. and Y.-W.L.; validation, J.-H.G., T.-L.C. and Y.-C.C.; writing—original draft, J.-H.G. All authors have read and agreed to the published version of the manuscript.

**Funding:** This research was funded by the Ministry of Science and Technology Taiwan, for financially supporting this research under Contract MOST 109-2221-E-157-001.

**Acknowledgments:** The authors would like to thank the Ministry of Science and Technology Taiwan for financially supporting this research under Contract MOST 109-2221-E-157-001.

**Conflicts of Interest:** The authors declare that they have no known competing financial interests or personal relationships that could have appeared to influence the work reported in this paper.

## Nomenclature

| | | |
|---|---|---|
| $C$ | Concentration | wt% |
| $F$ | Force | N/m |
| $K_w$ | Permeability | $m^2$ |
| $L$ | Substrate wetting length | mm |
| $L_w$ | Wick structure length | mm |
| $\dot{m}$ | Mass flow rate | g/s |
| $\Delta P_c$ | Wick Pressure difference | $Kg/cm^2$ |
| $\Delta P$ | Pressure difference | $Kg/cm^2$ |
| $Q$ | Total heat load | W |
| $R_{LHP}$ | Total thermal resistance | °C/W |
| $r_c$ | Effective radius | μm |
| $T$ | Fluid Temperature | °C |
| $T_c$ | Condenser temperature | °C |
| $T_{evap}$ | Evaporator temperature | °C |
| $T_w$ | Wick structure thickness | mm |
| $V_{pore}$ | Volume of pores in the wick | $m^3$ |
| $V_{total}$ | Volume of the entire wick | $m^3$ |
| $wt$ | Percent weight per volume | % |
| Alphabet | | |
| $\varepsilon$ | Porosity | % |
| $\mu$ | Viscosity coefficient | $Kg/cm^2 \cdot S$ |
| $\pi$ | pi | |
| $\rho$ | Density | $g/cm^3$ |
| $\sigma$ | Surface tension | mN/m |

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
