# Peer review of "The Study of Novel Self-Rewetting Fluid Application to Loop Heat Pipe"

_applsci, doi:10.3390/app12063121_

Round 1
Reviewer 1 Report
Dear authors,
Congratulations for the work you have done. It is an interesting subject.
I have some remarks:
You have abused in using “In this paper”. It is ok using 2-3 times, but no more than that.
Line 31 - it should “be strong.” Anyway, I don’t understand the last part of the sentence.
Line 38 – “solution, observable” what is the sense of observable?
Lines 44-45 – “self-rewetting fluids (Self-rewetting fluid)” should be “self-rewetting fluids (SRF)”
Line 53 “The theory is as follows: eq.(1)” – I do not understand this sentence.
Line 59 – correct “LHP” to “Loop Heat Pipe (LHP)”
Line 64 – you have there the word “Phenomenon” – it is single, what do you want to say by this?
Line 140 – correct to “an obvious”
Line 147 – delete “in the literature”, you have pointed this in the beginning of phrase
Line 150 – Replace “So, this paper tried to use “by “In this experiment it was used”
Correct caption of figure 2 : “Figure2.” replace by “Figure 2.”
The word “wick” in the caption of figure 3 should be “Wick”
In figure 3 you represent an arrow marked “Shock” pointing an image with some thunders. When you describe the preparation, you did not make any references to this in the text. What was the sintering atmosphere, if any?
Line 172 remove “In this paper”
Lines-172-174 – The phrase is ambiguous.
Line 176 – Replace “This paper using” by “In this experiment the loose filling method was used …”
Line 178 – remove “The sintered wick has good capillary force and other related parameters.” I do not think you have enough data in this stage of presentation to affirm this, maybe in the results section.
Line 184 – replace “will be” by “were”
Figure 4. I do not see where is connected to the system the pipe starting from water tank.
Line 201 – Remove “In this paper”
You said you used Archimedes Method to determine the porosity. The measurement was performed on the stand presented in figure 4. How was done?
Line 206 – replace “will” by “were”
Line 210 - Remove “In this paper”
Lines 215-216 – I do not understand the phares “And then consider …”
Lines 219-220 – remove “in this paper”
Lines 240-244 – The end of phrase is ambiguous
Line 261 Correct to “In figure 6 it can be observed …”
Please correct the caption of figure 7: Figure 7. (a) Ni wick, (b) Surface detail x50, (c) SEM image x1000
Enlarge the photos in figure 7, especially (a) and (b)
Line 280 – I do not see the internal structure as you mention here.
Line 282 – replace “porous” by “por”
I will suggest using a porosimeter (like mercury porosimeter) to determine the por size and distribution. Your SEM analyze was conducted only on the surface.
Line 286 – Reformulate the phrase.
You do not have “Conclusions”.
Chapter 4 is not representing “Discussions”, it is more like some “Conclusions”.
Remove lines 342, 349, 353, 358.
Best SRF Formulation: it is an ambiguous formulation.
Correct lines 398, 402, 407, 411, 413, 416, 417.
Author Response
Dear reviewer:
I am very grateful to review’s comments for the manuscript. We revised the manuscript in accordance with the reviewers’ comments, and carefully proof-read the manuscript to minimize typographical, grammatical, and bibliographical errors.
Attached is our description of the revision based on the reviewer's comments.

Reviewer 2 Report
1. The style is similar to the following paper.
"Study of self-rewetting fluid applied to loop heat pipe with PTFE wick".
2. Grammar edits needed for the sentences in the introduction paragraph 1. (lines 3 and 4) and some other sections in the paper.
Author Response
Dear reviewer:
I am very grateful to review’s comments for the manuscript. We revised the manuscript in accordance with the reviewers’ comments, and carefully proof-read the manuscript to minimize typographical, grammatical, and bibliographical errors.
Here below is our description on revision according to the reviewers’ comments.
Point 1: The style is similar to the following paper.
"Study of self-rethis manuscripttting fluid applied to loop heat pipe with PTFE wick".
Response 1: Although both are studies on self-rewetting fluid (SRF), the direction and focus of this paper is different from Wu et al. [11]. Wu et al. [11] focused on making PTFE wick available for loop heat pipe (LHP) through SRF , and reduce the heat leakage problem of LHP. The purpose of this paper is to apply the novel SRF to the conventional nickel wick to improve its heat transfer performance, and to confirm the novel SRF (2-Butanol 20%) as the best SRF with performance through experimental data.
Point 2: Grammar edits needed for the sentences in the introduction paragraph 1. (lines 3 and 4) and some other sections in the paper.
Response 2: The grammar in the article has been corrected.

Reviewer 3 Report
The novelty of this paper is unknown. Other related papers have been introduced. Therefore, the parts that differ from these papers should be clarified. I also wrote the results in Discussion, but there is no consideration. I think we should write a consideration such as why the case of 2-Butanol 20% aqueous solution is the most improved. The results are interesting, so I think they may be published in this magazine if they improve. Therefore, this paper is the major division.
Author Response
Dear reviewer:
I am very grateful to review’s comments for the manuscript. We revised the manuscript in accordance with the reviewers’ comments, and carefully proof-read the manuscript to minimize typographical, grammatical, and bibliographical errors.
Here below is our description on revision according to the reviewers’ comments.
Comment#1:
The novelty of this paper is unknown. Other related papers have been introduced. Therefore, the parts that differ from these papers should be clarified. I also wrote the results in Discussion, but there is no consideration. I think this manuscript should write a consideration such as why the case of 2-Butanol 20% aqueous solution is the most improved. The results are interesting, so I think they may be published in this magazine if they improve. Therefore, this paper is the major division.
Reply:
The innovation of this paper lies in the presentation of 2-Butanol 20%. 2-Butanol 20% has the best performance due to better initial wettability and a greater surface tension reversal gradient after 40°C, which allows for faster replenishment of the working fluid and improved performance.

Round 2
Reviewer 1 Report
Dear authors, most of the issues were corected.
Figure 4a, what is the role of filter, the conections for the filter don't show clear the circuit. Maybe you can make it smaller and see the entire filter and conections. In figure 4b, what means "graduate"?
Best regards,
Author Response
Dear reviewer:
I am very grateful to review’s comments for the manuscript. We revised the manuscript in accordance with the reviewers’ comments, and carefully proof-read the manuscript to minimize typographical, grammatical, and bibliographical errors.
Here below is our description on revision according to the reviewers’ comments.
Comment#1:
Dear authors, most of the issues were corected.
Figure 4a, what is the role of filter, the conections for the filter don't show clear the circuit. Maybe you can make it smaller and see the entire filter and conections. In figure 4b, what means "graduate"?
Reply:
The tube in Fig 4a should be filtered by Filter to allow the working fluid to be filtered by the filter, Fig 4a has been corrected.
In Fig 4b, graduate refers to beaker flask, but here it should use container, also corrected.

Reviewer 3 Report
The response to the question or comment was appropriate, so this paper is accepted.
Author Response
Thanks to reviewer's support and affirmation